# Identification of Genetic Variants Associated with Hereditary Thoracic Aortic Diseases (HTADs) Using Next Generation Sequencing (NGS) Technology and Genotype–Phenotype Correlations

**DOI:** 10.3390/ijms252011173

**Published:** 2024-10-17

**Authors:** Lăcrămioara Ionela Butnariu, Georgiana Russu, Alina-Costina Luca, Constantin Sandu, Laura Mihaela Trandafir, Ioana Vasiliu, Setalia Popa, Gabriela Ghiga, Laura Bălănescu, Elena Țarcă

**Affiliations:** 1Department of Medical Genetics, Faculty of Medicine, “Grigore T. Popa” University of Medicine and Pharmacy, 700115 Iași, Romania; 2Departament of Cardiology, Saint Mary’s Emergency Children Hospital, 700309 Iași, Romania; g_russu@yahoo.com (G.R.); alina.luca@umfiasi.ro (A.-C.L.); 3Department of Mother and Child, Faculty of Medicine, “Grigore T. Popa” University of Medicine and Pharmacy, 700115 Iași, Romania; laura.trandafir@umfiasi.ro (L.M.T.); gabriela.ghiga@umfiasi.ro (G.G.); 4Department of Medical Abilities, Faculty of Medicine, “Grigore T. Popa” University of Medicine and Pharmacy, 700115 Iași, Romania; sandu.v.constantin@umfiasi.ro; 5Department of Morphofunctional Sciences II, Grigore T. Popa University of Medicine and Pharmacy, 700115 Iași, Romania; ioana.vasiliu@umfiasi.ro; 6Department of Pediatric Surgery and Anaesthesia and Intensive Care, “Carol Davila” University of Medicine and Pharmacy, 020021 Bucharest, Romania; laura7balanescu@yahoo.com; 7Department of Surgery II—Pediatric Surgery, “Grigore T. Popa” University of Medicine and Pharmacy, 700115 Iași, Romania; tarca.elena@umfiasi.ro

**Keywords:** hereditary thoracic aorta diseases, aortic aneurysm, aortic dissection, genetic testing, Marfan syndrome, Loeys–Dietz syndrome, arterial tortuosity syndrome

## Abstract

Hereditary thoracic aorta diseases (HTADs) are a heterogeneous group of rare disorders whose major manifestation is represented by aneurysm and/or dissection frequently located at the level of the ascending thoracic aorta. The diseases have an insidious evolution and can be encountered as an isolated manifestation or can also be associated with systemic, extra-aortic manifestations (syndromic HTADs). Along with the development of molecular testing technologies, important progress has been made in deciphering the heterogeneous etiology of HTADs. The aim of this study is to identify the genetic variants associated with a group of patients who presented clinical signs suggestive of a syndromic form of HTAD. Genetic testing based on next-generation sequencing (NGS) technology was performed using a gene panel (Illumina TruSight Cardio Sequencing Panel) or whole exome sequencing (WES). In the majority of cases (8/10), de novo mutations in the *FBN1* gene were detected and correlated with the Marfan syndrome phenotype. In another case, a known mutation in the *TGFBR2* gene associated with Loeys–Dietz syndrome was detected. Two other pathogenic heterozygous variants (one de novo and the other a known mutation) in the *SLC2A10* gene (compound heterozygous genotype) were identified in a patient diagnosed with arterial tortuosity syndrome (ATORS). We presented the genotype–phenotype correlations, especially related to the clinical evolution, highlighting the particularities of each patient in a family context. We also emphasized the importance of genetic testing and patient monitoring to avoid acute aortic events.

## 1. Introduction

Hereditary thoracic aorta diseases (HTADs) are a heterogeneous group of rare pathologic conditions whose major manifestation is represented by aneurysm and/or dissection frequently located at the level of the ascending thoracic aorta. The diseases have an insidious evolution and can be encountered as an isolated manifestation (non-syndromic HTADs) or can also be associated with systemic, extra-aortic manifestations (syndromic HTADs) [1,2].

### 1.1. Epidemiology

Aortic aneurysms (AAs) and dissections are major conditions and represent leading causes of morbidity and mortality worldwide [3]. The incidence of thoracic aortic aneurysm (TAA) is estimated at 5 to 10 per 100,000 persons/year [4], and that of acute aortic dissection (AAD) at 5 to 30 cases/1 million people per year [5]. Estimating the incidence of TAA combined with AAD is difficult because many cases remain undiagnosed [4].

Weakening of the arterial wall is the main cause of AA and AAD associated with abnormal aortic hemodynamic profiles that predispose a patient to dilation of the thoracic aorta. The manifestation is usually silent, but it can precipitate dissection or rupture of the aorta with often fatal consequences [6]. It is estimated that about 95% of non-syndromic cases are silent, and only when AA reaches very large dimensions in the vast majority of cases, the first symptom is sudden death [7]. Approximately 2 to 7.3% of sudden cardiac deaths (SCD) in the general population are attributed to aortic ruptures and aortic dissections, which are considered real “silent killers” [7].

Aortic aneurysm (AA) is defined as a localized abnormal dilation of the aorta with a diameter at least 1.5 times the normal expected diameter [6]. Most frequently, AAs are located in the infrarenal abdominal aorta, followed by the ascending thoracic aorta. Thoracic aortic dissection (TAD) can also occur in the absence of aneurysmal dilatation (AA), being the result of the separation of the aortic media caused by extraluminal blood flow, which creates a false lumen in the layers of the aortic wall [7]. Rupture of the aorta causes a flow of blood from the inner orifice to the middle layer, causing the deformation of the wall and the expansion of the blood flow along the longitudinal axis [7].

Current strategies for assessing the future risk of aortic dissection or aortic rupture are primarily based on monitoring the diameter of the aorta (cardiac Z-score). However, this parameter alone is considered a poor predictor of risk, as many of the patients experienced dissection or rupture below the risk threshold [8,9].

Surgical repair of AA (root/ascending aorta) to prevent aortic dissection (AAD) is recommended at an aortic diameter of 5.0–5.5 cm. However, the majority of patients with dissecting aneurysms of the ascending aorta have diameters < 5.5 cm (AAD type A) or occur in the absence of significant aortic enlargement (AAD type B), indicating that more accurate predictors of AAD are needed [9].

### 1.2. Etiology of HTAD

In approximately 20–25% of cases of non-syndromic HTADs, the family history is positive for an aortic disease, which suggests the existence of a significant genetic etiological component [10]. In these cases, a clinical evaluation of the proband’s family members is mandatory to differentiate familial from sporadic cases [10].

The identification of a genetic cause of HTADs has a significant impact on the management of patients, as there is a correlation between the genetic defect, the clinical manifestations, and the evolution of the disease, as well as the treatment strategies. The genetic screening of family members would allow the early detection of asymptomatic carriers, for whom preventive measures are required, which also include ultrasound monitoring of the aorta dimensions (cardiac Z-score) [10,11].

In recent years, with the development of molecular technology that allowed extensive genetic testing (gene panels, whole exome sequencing—WES, or whole genome sequencing—WGS) using next-generation sequencing (NGS) technology, important steps have been taken in deciphering the heterogeneous genetic etiology both of syndromic and non-syndromic HTADs (Table 1).

Along with genetic factors, other risk factors for TAA include hypertension, smoking, hypercholesterolemia, male sex, advanced age (over 65 years), and the presence of a bicuspid aortic valve (Figure 1) [9,12,13].

#### Genetic Variants Associated with HTADs

An autosomal dominant (AD) monogenic mutation with high penetrance is detected in approximately 20–25% of families with HTAD [2,9].

Renard et al. [13] showed that 11 (*ACTA2*, *COL3A1*, *FBN1*, *MYH11*, *MYLK*, *SMAD3*, *TGFB2*, *TGFBR1*, *TGFBR2*, *PRKG1* and *LOX*) of the 53 candidate genes for HTADs that were analyzed are genes that cause HTADs [13]. In the case of patients with HTAD in whom mutations are identified in one of the eleven genes involved, clinical screening and genetic testing of family members are indicated in order to prevent the risk of sudden death and premature deaths of relatives carrying the same genetic variant [13].

Genetic risk correlates with the genetic variant detected. It is higher for extremely rare penetrant variants that cause disease in almost all carrier individuals and is lower for common variants found in the population that have low penetrance [13,14,15].

The research performed over time in patients with HTAD allowed the deciphering of the mechanisms and complex pathological processes at the level of the aortic wall aneurysm. Histopathologically, medial cystic necrosis with fragmentation and focal loss of elastic fibers occurs in TAA, accompanied by a decrease in vascular smooth muscle cells (VSMCs) and an increase in proteoglycans [16,17]. 

These changes can be associated with mutations in some genes (*MYH11*, *PRKG1*, *FOXE3*, *MLCK*, and *ACTA2*) that ensure the structural integrity of the aortic wall. These genes encode proteins (collagen, elastin, fibulins, and integrins) involved in VSMC adhesion and contraction, which underlines the importance of maintaining the elastin contractile unit [14] (Figure 1). Other genes (*FBN1*, *BGN*, *PLOD1*, *FNB2*, *ELN*, *LOX*, *EMILN1*, *COL1A1*, *COL3A1*, *COL5A1*, and *MFAP5*) are involved in extracellular matrix (ECM)-induced signaling pathways [14,18].

Clinical evaluations of familial TAAs may reveal the presence of associated systemic phenotypic manifestations involving the skeletal system or other connective tissue disorders (CTDs), which are commonly identified in patients with syndromic HTAD (e.g., Marfan syndrome or Loeys–Dietz syndrome) [13].

Genetic factors associated with Marfan syndrome (MS) and Loeys–Dietz syndrome (LDS) intervene in the TGF-β signaling pathway. Studies on *SMAD2*, *SMAD3*, *SMAD4*, *SMAD6*, *TGFRB1*, *TGFRB3*, *FLNA*, *ACTA2*, *MYH11*, *FBN2*, *BGN*, and *EMILN1* associated with syndromic HTADs have made a major contribution to determining the onset and progression of aortic aneurysm [18,19]. TGF-β signaling plays a critical role at the vascular level in blood vessel development, the regulation of contractile protein expression, cell proliferation and differentiation, and in homeostasis. These mechanisms are disrupted in all types of TAA that are associated with LDS [19,20].

### 1.3. Clinical Manifestations of Syndromic HTADs

*FBN1*-related MS manifests as a CTD with systemic manifestations, with a highly variable phenotypic spectrum, from specific manifestations of MS affecting only one or a few systems, to the severe neonatal form and rapidly progressive multiorgan disease [21].

The cardinal manifestations involve the cardiovascular and skeletal systems (pectus excavatum/carinatum, scoliosis, dolichostenomelia, arachnodactyly, and joint laxity), associated with ocular (ectopia lentis, retinal detachment, glaucoma and premature cataracts) and skin (laxity and skin striae) manifestations (Ghent criteria) [21].

Morbidity and early mortality of MS are correlated with cardiovascular complications, which include dilatation of the aorta at the sinuses of Valsalva, inducing aortic rupture, mitral valve prolapse (MVP) with or without regurgitation, tricuspid valve prolapse (TVP), proximal pulmonary artery dilatation, left ventricular dysfunction, and heart failure. The management of patients with MS requires periodic monitoring of the size of the aorta (cardiac Z-score) to avoid the occurrence of lethal complications [21].

In most cases of MS, *FBN1* missense variants are detected, which most frequently have a dominant-negative effect (DN-FBN1). Their consequences are a disorganized extracellular matrix (ECM) that includes both abnormal fibrillin-1 and normal proteins [22,23,24].

In 35% of MS cases, haploinsufficient *FBN1* (HI-*FBN1*) variants are identified and associated with reduced production of normal fibrillin-1 [22,23,25]. The presence of HI-*FBN1* mutations in MS patients correlates with a faster rate of aortic root and ascending aorta dilatation and an increased risk of aortic dissection and death, but a better response to losartan therapy compared to patients with MS in whom a DN-*FBN1* variant is identified [23,24,25,26,27].

Loeys–Dietz syndrome (LDS, ORPHA:60030) is an AD syndromic aortopathy characterized by the association between arterial tortuosity with aneurysm/dissection of the ascending aorta, including the distal aorta, and craniofacial features (hypertelorism, strabismus, bifid uvula/cleft palate, and craniosynostosis) [28]. The clinical picture also includes skeletal manifestations (pectus excavatum or pectus carinatum, scoliosis, joint laxity, arachnodactyly, talipes equinovarus, and cervical spine malformation and/or instability) and cutaneous findings (velvety and translucent skin, easy bruising, and dystrophic scars). The LDS phenotype is associated frequently with loss-of-function (LOF) mutations in six genes (*TGFBR1*, *TGFBR2*, *SMAD2*, *SMAD3*, *TGFB2*, and *TGFB3*) involved in the TGF-β signaling pathway [28].

Vascular Ehlers–Danlos syndrome (vEDS) is an AD form of syndromic HTAD and is frequently caused by heterozygous mutations in the *COL3A1* gene and, more rarely, in the *COL1A1* gene. vEDS is manifested by arterial, uterine, or intestinal ruptures; translucent skin with visible circulation; ecchymoses from minor trauma; skeletal anomalies; and characteristic facial dysmorphism. The prognosis is more severe in males compared to females, and management includes periodic arterial screening [29]. The clinical phenotype and prognosis of vEDS correlate with the type of *COL3A1* variant identified. Approximately 5% of *COL3A1* variants are LOF mutations that cause haploinsufficiency. The *COL3A1* “null” variant is associated with reduced penetrance and exclusively vascular events. In these patients, complications appear 15 years later and the life expectancy is better, with fewer obstetric and intestinal complications compared to those caused by other *COL3A1* pathogenic variants (missense and splicing variants) [29,30,31].

Pathogenic variants of the *SLC2A10* gene, which encodes the GLUT10 protein that regulates the TGF-β signaling pathway, cause arterial tortuosity syndrome (ATORS; OMIM #208050, ORPHA:3342). ATORS is associated with the elongation, tortuosity, and aneurysms of the large and medium arteries with craniofacial dysmorphism (long face, hypertelorism, downslanting palpebral fissures, beaked nose, sagging cheeks, a high palate, and micrognathia), skin (soft and hyperextensible skin, cutis laxa, and hernias), skeletal (joint hypermobility and congenital contractures), and ocular (keratoconus) manifestations and generalized hypotonia. The cardiovascular abnormalities include pulmonary and aortic stenosis, coarctation of the aorta, abnormal implantation of the aortic branches, and an increased risk of ischemic events [32].

Impaired expression of matrix metalloproteinases (MMPs) and their inhibitors [tissue inhibitors of metalloproteinases (TIMPs)] causes proteolysis of the medial layer of the aorta. These mechanisms seem to be involved in the occurrence of TAA in patients with bicuspid aortic valve (BAV). Approximately 40% of patients with BAV are predisposed to dilatation of the ascending aorta, which is considered an independent risk factor for TAA [33,34].

BAV is inherited as an AD trait with incomplete penetrance and variable expressivity caused by the interactions of multiple genes [35]. Taking into account the association between BAV and TAA, the role of genetic variants (e.g., *NOTCH1*, *ROBO4*, *SMAD6*, *ELN*, *FBN1*, *ACTA2*, and *LOX* variants) associated with syndromic TAA in the etiology of BAV was discussed [36,37].

The hypothesis of the involvement of metalloproteases (MMPs) was discussed, starting from the observation of significant increases in MMP-2 levels associated with a reduction in TIMP-1 levels in patients with BAV/TAA, compared to subjects with TAA who presented a normal tricuspid valve; increased expression of MMP-2 and MMP-9 in patients with BAV could explain the higher prevalence of TAA in them [38,39]. A case–control study conducted in 2018 reported a close association between serum levels of the MMP-9 isoform and AA, suggesting that MMP-9 could be a marker especially for TAA [40].

However, research has shown that these markers are not specific for the TAA, as their increased levels can be detected in abdominal aortic aneurysm (AAA) (MMP-3, MMP-8, MMP-12, and TIMP-3), kidney diseases (TIMP-2) or cancers (MMP-9, TIMP-1, and TIMP-2) [41].

Due to the onset at young ages and the devastating complications, TAD is a major cause of death in the young population. Taking into account the silent, asymptomatic evolution of TAAs, their early detection and careful monitoring, which includes echocardiographic measurements of the aortic diameter, are very important to avoid severe complications. Sometimes TAAs are discovered accidentally during a routine clinical examination with the help of imaging techniques (echocardiography, magnetic resonance angiography (MRA), and computed tomographic angiography (CTA)) [6].

In recent years, research studies have considered the development of some tools (for example, biochemical markers) a priority, which, together with imaging methods, would allow the improvement of risk assessment and management in patients with aortopathy, including HTAD. 

Extensive genetic testing (gene panels, WES, or WGS) allowed the heterogeneous etiology of HTADs (syndromic and non-syndromic) to be deciphered (Table 1). In addition, the identification of new candidate genes for HTADs created the premise of new therapeutic targets. Molecular genetic testing of patients with HTAD is important because it allows the identification of the genetic variants involved, predicts the likely clinical course, and can influence the patient’s treatment [10].

### 1.4. Genetic Counseling

Genetic counseling is a major step in the management of patients with HTAD. The risk of disease recurrence in first-degree relatives of the proband varies between 25 and 50%, depending on the type of autosomal dominant or recessive inheritance of the diseases [1]. Genetic screening of family members allows identification of other individuals at increased risk for HTADs. In addition, the early identification of asymptomatic carriers (individuals who have inherited the mutations, but do not have phenotypic manifestations) allows preventive measures for TAA and TAD through individualized aortic surveillance [11]. 

The aim of our study is to identify genetic variants associated with hereditary thoracic aortic diseases (HTADs), which are characterized by an increased risk of aortic dilation and dissection, in a group of patients diagnosed with syndromic forms of the disease.

We also present the genotype–phenotype correlations and highlight the particularities of each patient in the context of his family, correlated with the identified genetic variant, especially variants related to the phenotypic manifestations, management, and personalized treatment.

## 2. Results

Extended genetic testing (gene panel that included genes for aortopathies and WES) of ten patients who presented clinical signs specific to some syndromes that are associated with HTADs identified a likely pathogenic (LP) missense mutation in the *FBN1* gene (*FBN1* c.7343G>A, *FBN1* c.5743C>T, *FBN1* c.6806T>C and *FBN1* c.3023G>A) in eight of them, and the result correlated with a clinical suspicion of Marfan syndrome (MS) (Table 2). A frameshift LP mutation in the *FBN1* gene (*FBN1* c.7168dup) was detected in a single patient (P07) with MS (Table 2).

In one patient (P09, P.N.) a pathogenic missense variant in the *TGFBR2* gene was identified and correlated with LDS. In another patient (P10, D.N.B), two heterozygous mutations (one missense, pathogenic variant and the other a pathogenic nonsense variant, most likely in trans) were detected in the *SLC2A10* gene associated with ATORS (Figure 2).

Four of the five mutations detected in patients with Marfan syndrome (MS) were new mutations that were not reported in the literature. The first three patients (P01, P02, and P03) came from the same sibling, and the fourth (P04) was their first cousin from the paternal line (Figure 3). 

In the other four patients diagnosed with MS, who came from different, unrelated families, new variants of *FBN1* were also detected (Table 1, Figure 2) The family history of aortic disease (aortic dilation and/or dissection) was positive in six of the ten analyzed patients (P01, P02, P03, P04, P05, and P06), all being diagnosed with MS. The clinical data of the ten patients included in the study are presented in Table 3.

The age of the patients at the time of diagnosis varied between 3 months and 20 years, and four of the ten analyzed patients were female (P02, P04, and P07) and six were male (P01, P03, P05, P06, P08, P09, and P10) (Table 3). All patients presented cardiac manifestations, the most frequent being mitral valve prolapse (MVP) (P01, P02, P03, P04, P05, P06, P07, and P08). Four of the patients had aortic bulb (AB) ectasia (P03, P06, P07, and P08) and two others patients had dilatation of the coronary sinus (CS) (P01 and P02) (Table 3). 

In three patients with AB ectasia (P06, P07, and P08) a cardiac Z-score > 2 was detected, and surgical intervention was discussed to avoid potentially lethal complications. Skeletal manifestations (tall stature, arachnodactyly, pes planus, and pectus excavatum) were present in all patients diagnosed with MS. Myopia was detected in five of the eight patients with MS (P02, P03, P04, P05, and P07), and ectopia lentis associated with congenital cataract was present in only one patient (P08) with MS. 

All patients presented specific manifestations of connective tissue disorders (CTDs): three of the patients presented skin striae (P01, P05, and P07), another four patients with inguinal/inguinoscrotal hernia (P01, P04, and P08 with MS and P10 with ATORS), and only one patient with MS presented joint laxity (P06) (Table 3). Other associated clinical manifestations were craniofacial dysmorphism present in all patients, and macrocephaly and sleep disorders detected in patient P03 diagnosed with MS (Table 3).

## 3. Discussion

In most patients with HTAD included in the study (eight of the ten patients), molecular genetic testing (gene panel or WES) identified a pathogenic mutation in the *FBN1* gene. The correlation between the detected genetic variant (genotype) and the severity of aortic events (phenotype) was investigated in all patients. 

Marfan syndrome (MS) is a pleiotropic connective tissue disorder with variable expressivity inherited as an autosomal dominant trait and is caused in most cases by a mutation in the *FBN1* gene located on chromosome 15q21.1, encoding fibrillin-1 [12,21,42].

According to one of the most extensive databases with mutations in the *FBN1* gene (The International *FBN1* Universal Mutation Database, UMD-*FBN1*), 1847 different mutations are reported in 3044 DNA samples [http://www.umd.be/FBN1/, accessed on 23 July 2024] [43]. Mutations are distributed over the entire length of the *FBN1* gene, and only 12% of them are recurrent [44,45].

The number of mutations is probably much higher because not all variants identified worldwide are reported and entered into the database [44]. In approximately 25% of cases, de novo mutations are detected, while 75% of patients have a positive family history [21]. In the case of *FBN1* pathogenic variants, the reported penetrance is 100%; thus, people who inherit a pathogenic *FBN1* variant from a parent will show symptoms specific to MS, without being able to predict their severity [21]. A parent with somatic and/or germline mosaicism for a pathogenic *FBN1* variant may have reduced phenotypic manifestations. Advanced paternal age (36 vs. 29 years) at conception contributes to sporadic cases of MS [46,47]. 

Several types of mutations in the *FBN1* gene are described in patients with MS [21,48]. In approximately two-thirds of the cases, *FBN1* missense mutations are identified, which usually replace the cysteine residues that form disulfide bonds within one of the calcium binding epidermal growth factor-like (cbEGF) domains or 8-cysteine (8-Cys) containing domains, but there are also frequent mutations that create new cysteine residues in these modules [44,49,50]. About 25% of missense mutations affect modules other than the cbEGF domains of fibrillin-1 [49,50].

Other types of mutations are in-frame deletions due to exon-skipping/splice-site mutations [51] or genomic deletions [51,52]. Mutations that most frequently affect the canonical splice sequences at exon/intron boundaries are detected in approximately 10–15% of cases. Many *FBN1* splice-site mutations cause in-frame exon skipping with the absence of the entire cbEGF domain in mutant fibrillin-1 [53].

Small insertions, deletions, and duplications are detected in 10–15% of cases and cause a premature stop codon (PTC) and the synthesis of a truncated protein [54]. Deletions of the entire gene are extremely rare, and larger rearrangements (deletions or insertions) are described in a small proportion of patients with MS [55,56,57].

In about 25% of cases, frameshift or nonsense mutations in the *FBN1* leading to premature termination codons (PTC) are identified [58,59,60].

Taking into account the genetic heterogeneity present in patients with MS, the screening of *FBN1* mutations should include both common techniques for the detection of exonic mutations, as well as methods for the detection of large deletions/insertions (Multiplex Ligation-dependent Probe Amplification, MLPA) and even for deep intronic mutations (whole genome sequencing, WGS) [47,61].

### 3.1. Genotype–Phenotype Correlations in Marfan Syndrome

The most severe manifestations of MS are associated with multiexon deletions [62,63]. The best genotype–phenotype correlation that has been reported is the association of mutations in the middle region of the *FBN1* gene (deletions of exons 24–32) that correlate both with severe forms of neonatal MS and with other severe forms of the disease [25,64,65,66]. The presence of a mutation in this region of the *FBN1* gene seems to be the best risk indicator for early-onset aortic disease in the absence of neonatal MS [44,48,67].

Schrijver et al. [50] showed that mutations that change a cysteine residue or a cbEGF-like domain are associated with the classic form of MS [50]. Patients with non-cysteine missense mutation in the *FBN1* gene have more severe cardiovascular (AA and MVP) and skeletal manifestations than patients with missense mutations involving cysteine [https://dare.uva.nl/search?identifier=420cb2ef-5ecf-4194-980e-8a7f5873aadc, accessed on 24 July 2024] [50,67].

Baudhuin et al. [27] identified the presence of protein-truncating or splicing variants in most patients who had an early aortic event [27]. Aortic dissection (AAD) as well as prophylactic surgery occurred at a younger age (mean age, 29 years) in patients with a protein-truncating or splicing genetic variant compared to those with missense variants (mean age, 51 years) [27].

In a study that included 1009 probands with pathogenic *FBN1* mutations (320 under 18 years of age), Faivre et al. [68] shows that nonsense, frameshift, and splicing mutations in exons 24–32 were detected only in 27.8% of patients compared to all other exons (44.4%), and nonsense mutations were not present in any patients with neonatal MS [68].

Although global data indicate that a higher risk for early aortic events correlates with truncating and splicing mutations in exons 24–32, the phenotypic severity cannot be accurately predicted based only on detected genomic variants [21,27,44,64,68].

The effects of mutations in the *FBN1* gene can be influenced by many factors that regulate both gene expression (transcription and mRNA translation) and the post-translational processing of fibrillin-1 (glycosylation, signal peptide cleavage, and carboxyl-terminal processing), including multimerization, incorporation in the extracellular matrix, and then its degradation. All these factors contribute to the phenotypic heterogeneity present in patients with MS [64].

In the first four patients with MS, coming from the same family, we found variable expressivity correlated with the *FBN1* c.7343G>A, p.(Cys2448Tyr) variant detected by molecular testing (Illumina TruSight Cardio Sequencing Panel).

The first three patients (P01, P02, and P03) came from the same sibling and were diagnosed with MS following the sudden death of their father due to the rupture of an aneurysm of the descending aorta. We mentioned that he had not previously been diagnosed with MS, and the diagnosis of aortopathy was pathologically confirmed after his death. 

The family anamnesis revealed the presence of other people in the paternal line who were most likely possibly affected but were not genetically tested: the paternal grandmother and a sister and a brother of the father (who was operated on for a giant aneurysm of the ascending aorta) (Figure 3). All of them had a marfanoid phenotype (tall stature, pectus excavatum, arachnodactyly, and ocular and cardiovascular anomalies).

The first patient (P01, F.B, aged 12 years) had a tall stature (+2.72 SD), facial dysmorphism, arachnodactyly, positive wrist and thumb signs, pes planus, and severe pectus excavatum that later required surgical correction. Cardiac ultrasound revealed the presence of an anterior MVP, mitral regurgitation, dilated coronary sinus (CS), and a cardiac Z-score < 2. Other specific manifestations of connective tissue disorders (CTDs) were skin striae and inguinal hernia (Table 2).

The second patient (P02, D.A.B, aged 9 years) presented a phenotype that was associated with a tall stature (+2.3 SD), macrocephaly (+2.15 SD), craniofacial dysmorphism (left palpebral ptosis), pectus excavatum, thoracic kyphosis, arachnodactyly, pes planus, and positive wrist and thumb signs. Cardiac ultrasound revealed the presence of anterior MVP, mild mitral regurgitation, and dilated coronary sinus (CS) (cardiac Z-score < 2) (Table 2).

The third patient (P03, M.B, aged 7 years) presented discrete facial dysmorphism, pectus excavatum, slight thoracic asymmetry, arachnodactyly, and pes planus. Cardiac ultrasound revealed anterior MVP, mild mitral regurgitation, and a cardiac Z-score < 2. Ectopia lentis was absent and psychomotor development was normal in all three brothers.

The fourth patient (P04, L.D.B) (first cousin in the paternal line of the first three patients) was aged 12 years at the time of diagnosis and had a tall stature, skeletal anomalies (arachnodactyly, pectus carinatum, cavus foot, positive wrist and thumb signs, severe myopia, and an inguinal hernia (operated) (Table 3). Psychomotor development was normal. At the time of the initial evaluation, the cardiac ultrasound did not reveal the presence of aortic ectasia or dilatation, and the orthopedic examination revealed an external tibial torsion (bilateral). The family anamnesis revealed that her father was diagnosed with a giant aneurysm of the ascending aorta, and the paternal uncle (father of the first three patients) died due to the rupture of an aneurysm of the descending aorta (Figure 3).

Through a sequencing analysis of a panel of 174 genes with the Illumina TruSight Cardio Sequencing Panel kit, a heterozygous missense variant c.7343G>A was detected in the *FBN1* gene located on chromosome 15q21.1. This *FBN1* genetic variant affects a highly conserved nucleotide. The variant NC 000015.9:g.48717676C>T NM_000138.4:c.7343G>A NP_000129.3: p.(Cisl448Tyr) is a cytosine-to-thymine substitution at the DNA level in exon 60/66 a1 of the *FBN1* gene, which caused the replacement of a cysteine residue with tyrosine in the protein structure at position 2448/2872. This type of mutation causes only a change in the amino acid in the protein structure without affecting its total number and subsequent sequence of amino acids (missense mutation). Among the 1230 known variants of the *FBN1* gene with an effect, 1188 (96.6%) are reported to be pathogenic. [http://www.umd.be/FBN1/, accessed on 23 July 2024] [43]. In addition, this variant is located in a “hot spot” region of the gene with a length of 61 base pairs, which contains seven pathogenic variants out of ten variants with known significance. An alternative variant, chr15:48717677 AWG c.7342T>C, p.(Cys2448Arg), is classified as “pathogenic” in the ClinVar database [https://www.ncbi.nlm.nih.gov/clinvar/, accessed on 26 July 2024] [69]. This mutation is very rare, with no frequency reported in the gnomAD Exomes or gnomAD Genomes databases, which have good coverage in this region, indicating that it is not a common benign variant in these populations [https://gnomad.broadinstitute.org/, (accessed on 1 April 2024)] [70]. The effect of certain mutations can be estimated by bioinformatic methods. These predictive scores target the qualitative impact of the mutation on protein function based on amino acid homology and the chemical structure. For the variant identified in our patients, the nine scores used (ANN, ElGEN, PATHMM-MKL, M-CAP, MVP, MutationTaster, PrimateAI, REVEL, and SIFT) placed it in the “affected/harmful/pathogenic” category. All this evidence is limited but suggestive of pathogenicity, which places the mutation in the category of “likely pathogenic variant with clinical significance.”

The fifth patient (P05, C.G, male, aged 9 years), who was diagnosed with MS, presented facial dysmorphism (downslanting palpebral fissures, left palpebral ptosis, dental malposition, deviated nasal septum, and prominent columella) associated with a tall stature (+2.1 SD), arachnodactyly, positive wrist and thumb signs, pectus excavatum, pes planus, skin striae, myopia, accessory spleen, and borderline intellectual functioning (IQ = 68) (Table 3). The family anamnesis was positive for aortic disease (the patient’s father died due to thoracic aorta dissection, but was not genetically tested). Also, three of the patient’s sisters, a paternal uncle, and the paternal grandfather presented a specific MS phenotype, but they were not genetically tested (Figure 3).

Molecular genetic testing (Illumina TruSight Cardio Sequencing Panel) identified a heterozygous missense mutation c.5743C>T in the *FBN1* gene. This variant affects a highly conserved nucleotide, leading to an amino acid change from Arg to Cys at codon 1915. Fibrillin-1 (FBN1) contains 47 epidermal growth factor (EGF)-like domains characterized by six conserved cysteine residues. A cysteine alteration in one of these domains could disrupt disulfide bonding and may affect the secondary or tertiary structure, or affect fibrillin-1 interactions, which is a common cause of MS. In silico prediction models* (11/13) predict that this missense variant has a deleterious effect on the protein, but no functional studies have been published. This variant has not been reported in large populations (GnomAD), indicating that it is not a common benign variant in these populations. Also, this variant has not been reported in patients with MS in the literature. For these reasons, the *FBN1* c.5743C>T variant was classified as likely pathogenic according to the American College of Medical Genetics and Genomics (ACMG) guidelines.

The sixth patient (P06, D.P.N, aged 6 years), presented a phenotype characteristic of MS with skeletal manifestations (tall stature, +2 DS, arachnodactyly, ligamentous laxity, pectus excavatum that required surgical intervention, pes planus, and lumbar lordosis) associated with MVP, mitral valve insufficiency (MI), aortic bulb (AB) ectasia, and a cardiac-Z score > 2 (3.55). The family history revealed that a sister, a brother, and the patient’s father were diagnosed with MS based on the clinical picture (but have not been genetically tested). In addition, the patient’s father was diagnosed with ascending aorta aneurysm (operated). In this case, molecular genetic testing (Illumina TruSight Cardio Sequencing Panel) detected a missense heterozygous mutation c.6806T>C, p.(Ile2269Thr) in the *FBN1* gene. The c.6806T>C, p.(Ile2269Thr) variant (rs193922228) has been reported in multiple individuals affected with MS or *FBN1*-related disorders (PMID:18435798, PMID: 17657824, PMID: 31098894, PMID: 25907466, PMID: 29848614, PMID: 19159394, and PMID: 19293843) [23,71,72,73,74,75,76] and is classified as pathogenic/likely pathogenic in ClinVar (variant ID 36107) [ClinVar database, available online at https://www.ncbi.nlm.nih.gov/clinvar/, (accessed on 26 July 2024)] [69].

This variant is absent from the general population databases (1000 Genomes Project, Exome Variant Server, and Genome Aggregation Database), indicating that it is not a common polymorphism) [Genome Aggregation Database, available online at https://gnomad.broadinstitute.org/, (accessed on 26 July 2024)] [70]. The isoleucine at codon 2269 is a highly conserved residue located within the cbEGF domain. Based on the above information, this variant is likely pathogenic. 

In the case of the seventh patient (P07, A.C), a female, the initial clinical evaluation at the age of 3 months revealed the presence of arachnodactyly associated with MVP. At the last evaluation at the age of 20 years, the patient presented skeletal manifestations (arachnodactyly, positive wrist and thumb signs, pectus carinatum, kyphoscoliosis, and pes planus), cardiac abnormalities (prolapse of both mitral valves, mitral regurgitation, AB ectasia, and a cardiac Z-score of 5.24) associated with myopia, skin striae, and facial dysmorphism (Table 3). The family history was negative, and no other affected persons were identified in the patient’s family (Figure 3).

By testing a gene panel (Illumina TruSight Cardio Sequencing Panel), a heterozygous frameshift mutation c.7180C>T, p.(Cys2390LeufsTer16) in the *FBN1* gene was identified. The variant found has not been reported in other patients with MS in the literature and is not reported in large population databases (GnomAD), but is thought to have a deleterious effect on the encoded protein. For these reasons, the *FBN1* c.7180C>T variant was classified as likely pathogenic according to the ACMG guidelines.

The eighth patient (P08, S.B., male) was diagnosed with MS at the age of 8 years. In his case, the clinical picture included skeletal manifestations (tall stature, arachnodactyly, positive wrist and thumb signs, and pes planus) associated with cardiovascular anomalies (hypertension, anterior MVP, mitral and aortic regurgitation, BAV, AB ectasia, and a cardiac Z-score of 3.2), ocular manifestations (congenital cataract and bilateral ectopia lentis) and inguinal–scrotal hernia. In addition, the patient had obesity, hypertriglyceridemia and borderline intellect (Table 3). The family history was negative. Molecular genetic testing (Illumina TruSight Cardio Sequencing Panel) identified a heterozygous missense mutation c.3020G>A, p.(Cys1008Tyr) in the *FBN1* gene. Although the identified variant has not been reported in the literature and is not present in the population databases, the in silico prediction models classify it (7/7) as having a harmful effect on the protein, CADD score = 33. For these reasons, the *FBN1* c.3020G>A, p.(Cys1008Tyr) variant was classified as likely pathogenic according to the ACMG guidelines.

Four of the five mutations detected in patients with MS (P01, P02, P03, P04, P05, P06, and P08) were de novo missense variants, while in one of the patients (P07, A.C.), a heterozygous frameshift mutation c.7180C>T, p.(Cys2390LeufsTer16) in the *FBN1* gene was identified.

Three of the detected missense mutations affect conserved cysteine residues by replacing cysteine with another amino acid in the protein structure (p.(Cys2448Tyr), p.(Arg1915Cys), p.(Cys1008Tyr), and in only one case, a highly conserved isoleucine residue located within the cbEGF domain (p.(Ile2269Thr) was affected. Although the number of patients included in our study is relatively small, we believe that we can say that the results obtained are consistent with those in the literature. Thus, the data reported so far reveal that in more than two-thirds of the patients with MS, missense mutations are detected, of which only 25% affect domains other than the cbEGF domains of fibrillin-1, in these patients, the cardiovascular and skeletal damage is much more severe. At the time of evaluation, only two of the patients with missense mutations (P06, age 6 years, and P08, age 8 years) presented AB ectasia and a cardiac Z-score > 2, and were considered for prophylactic aortic surgery.

Also, patient P07 (A.C, age 20 years) diagnosed with a heterozygous frameshift mutation c.7180C>T,p.(Cys2390LeufsTer16) in the FBN1) presented severe cardiovascular manifestations (MVP with annuloplasty, AB ectasia, and a cardiac Z- score of 5.24), and surgical intervention was indicated to avoid potentially lethal complications.

One of the patients (P09, P.D, female, age 4 years 10 months) was diagnosed with LSD. The clinical examination revealed the presence of connective tissue abnormalities (joint hypermobility and skin laxity, pectus carinatum, pes planus, and scoliosis) associated with cardiovascular anomalies (patent ductus arteriosus), low weight, and craniofacial dysmorphism (facial asymmetry, micrognathism, and low-set ears). There were no other affected people in the family. DNA sequencing analysis using the Blueprint Genetics (BpG) Whole Exome Plus identified a heterozygous missense variant c.1582C>T, p.(Arg528Cys) in the *TGFBR2* gene. This variant is absent in gnomAD, a large reference population database (n > 120,000 exomes and >15,000 genomes) that aims to exclude individuals with severe pediatric diseases. The variant results in the substitution of arginine (Arg) with cysteine (Cys) at protein position 528. The affected residue, located in the kinase domain of TGFBR2, is highly conserved among species, suggesting that substitutions at this position may not be tolerated (PMID 15731757) [77].

There is a large physicochemical difference between arginine and cysteine. Therefore, this variant is considered a radical substitution (Grantham score = 180); arginine and cysteine differ in polarity, charge, size, and/or other properties, and this variant is more likely to impact the secondary structure. The variant is predicted to be deleterious by all in silico tools utilized. The *TGFBR2* c.1582C>T, p.(Arg528Cys) variant (also reported as *TGFBR2* c.1657C>T, p.(Arg553Cys) in the literature) has been published in the literature and reported in the clinical variation databases ClinVar ID 12512, UMD-TGFBR2, and HGMD. The *TGFBR2* c.1582C>T, p.(Arg528Cys) mutation has been reported in at least seven patients with LDS (PMID: 16928994, PMID: 20144264, PMID: 17330129, and PMID: 18781618) [78,79,80,81].

LeMaire et al. [80] reported the variant in a 24-year-old male diagnosed with MS at 9 years of age with additional clinical features suggestive of LDS and a history of eight cardiovascular procedures to treat rapidly progressive aneurysms, dissection, and tortuous vascular disease (PMID 17330129) [80].

Stheneur et al. [81] identified the variant in a newborn with features of LDS, including bifid uvula, hypertelorism, and major cardiac involvement (PMID 18781618) [81]. 

Jamsheer et al. [82] identified the same mutation in a 2-year-old Polish patient who presented with typical manifestations of LDS syndrome that included craniofacial dysmorphism (craniosynostosis, cleft palate, and hypertelorism), arachnodactyly, camptodactyly, scoliosis, joint hypermobility, skin laxity, and umbilical hernia. Other associated findings were mild dilatation of the ascending aorta and a tortuous course of the left internal carotid artery (PMID 19875893) [82].

In addition, the *TGFBR2* c.1582C>T, p.(Arg528Cys) variant was also identified in a child with LDS including early onset aortic dilatation whose mother died soon after delivery due to aortic dissection, as well as in a young male with a clinical suspicion of LDS and family history of pectus excavatum (BpG, unpublished observations). Functional studies have shown that the variant results in decreased protein expression and has a dominant-negative effect on TGF-beta-induced Smad and ERK signaling pathways (PMID 21098638) [83]. Also, three other variants that are associated with LDS have been identified at the same amino acid codon. These are changes of arginine to histidine, proline, and serine. 

Grantham scores for the changes vary from conservative (Arg528His, Grantham score < 50), to radical (Arg528Cys, Arg528Ser, and Arg528Pro, Grantham scores >100), and the identified p.(Arg528Cys) variant is the most radical change (180 vs. 110 and 103) [OMIM, Clin VarDatabase] [12,69]. Mutations in the same residue and in nearby residues have been reported in association with LDS (Human Gene Mutation Database, HGMD), further supporting the functional importance of this residue and this region of the protein. [Human Gene Mutation Database (HGMD) database, available online at https://digitalinsights.qiagen.com/products-overview/clinical-insights-portfolio/human-gene-mutation-database/ (accessed on 23 August 2024)] [84].

The last patient (P10, D.N.B., male) was initially evaluated at the age of 3 months and then re-evaluated at the age of 1 year, and he presented skin laxity, joint hypermobility, inguinal–scrotal hernia, congenital clubfoot, and craniofacial dysmorphism associated with cardiovascular anomalies (hypertrophic cardiomyopathy, predominantly septal, aortic coarctation, and aortic regurgitation). Aortography revealed significant tortuosity of the aorta along its entire length, as well as of all the large arteries emerging from it, raising the suspicion of the diagnosis of arterial tortuosity syndrome (ATORS).

Genetic testing (Illumina TruSight Cardio Sequencing Panel) identified two mutations (compound heterozygous status) in the *SLC2A10* gene located on chromosome 20q13.12 (Table 2). The first is a heterozygous missense variant c.394C>T, p.(Arg132Trp) in the *SLC2A10* gene affecting a well-conserved nucleotide. This mutation replaces arginine, which is basic and polar, with tryptophan, which is neutral and slightly polar, at codon 132 of the SLC2A10 protein (p.(Arg132Trp). The variant is present at a low frequency in large populations in databases (rs121908173, gnomAD 0.007%), as well as in ClinVar (variation ID: 4590). The missense variant was reported in individuals with autosomal recessive ATORS (PMID: 17935213, 28726533, 29543232) [85,86,87]. In at least one of the patients, this variant was in trans (on the opposite chromosome) with another pathogenic variant (compound heterozygous status) [85,88]. 

This amino acid position is highly conserved in available vertebrate species. In addition, this alteration is predicted to be deleterious by an in silico analysis. Based on the supporting evidence, this alteration is interpreted as a disease-causing mutation (pathogenic variant). The second detected heterozygous mutation c.509G>A, p.(Trp170Ter) in the *SLC2A10* gene is a new stop-gain (nonsense) variant, which has not been reported in the literature. Other nonsense variants are reported to be pathogenic, representing a cause of ATORS. In silico prediction models (LRT and MutationTaster) predict that the *SLC2A10* c.509G>A, p.(Trp170Ter) variant has a deleterious effect on the protein. The variant has not been reported in large populations nor in other patients with ATORS in the literature. For these reasons, the *SLC2A10* c.509G>A variant was classified as pathogenic according to the ACMG guidelines. In the performed sequencing, four of the reads contained both variants and the results suggested that they are positioned in trans (on different chromosomes), but for certainty, testing of the patient’s parents will be necessary.

### 3.2. Are Genetic Factors the Cause of Faster Aortic Dilatation? The Importance of Genetic Testing and Genetic Counseling in Syndromic Forms of HTADs

Thoracic aortic aneurysm (TAA) is a potentially fatal disease, and the size of the aneurysm is correlated with the risk of aortic dissection and sudden death [10]. More than 40% of cases are hereditary, and the genetic etiology is heterogeneous [89]. Data from the literature highlight that genetic factors are associated with an early aortic dilatation and an increased risk of aortic dissection. Family history is an important aspect in the evaluation of patients with TAA/aortic dissection, distinguishing syndromic forms from isolated ones. Approximately 15% of TAA patients have an affected first-degree relative, and segregation studies of these families have suggested a major genetic component [14]. The identification of individuals at increased risk for a syndromic form of TAA from the patients’ families contributes to taking effective preventive measures that include continuous cardiac monitoring using clinical examinations and aortic imaging to avoid severe lethal complications, including sudden unexpected death [4,14].

Different studies have shown that aortic dissections occur at a younger age and at a smaller aneurysm diameter in patients with the inherited forms compared to sporadic HTADs [90]. Molecular genetic testing (next-generation sequencing) of a single gene/gene panel or WES plays an important role in the diagnosis of syndromic forms of TAA, because the genetic defect influences the location of the aneurysm and correlates with the risk of aortic dissection [90,91]. 

In the 2022 American College of Cardiology (ACC)/American Heart Association (AHA) Guideline for the Diagnosis and Management of Aortic Disease [4], genetic testing is recommended for individuals with syndromic features, a family history of TAD, and/or a young age of disease onset [4]. Imaging of the thoracic aorta (echocardiography, CTA, or MRA) is recommended for first-degree relatives of all individuals with TAD, regardless of the age of onset, to detect asymptomatic TAAs. In the case of identification of a pathogenic/likely pathogenic variant, genetic testing and aortic imaging of biological relatives at risk are recommended [4]. When genetic testing is negative or variants of unknown significance (VUS) are identified, aortic imaging screening is recommended in first-degree relatives of the proband. Also, aortic imaging screening of first-degree relatives is also recommended in the case of patients with aortic root/ascending aorta aneurysms or aortic dissection, in the absence of a known family history of TAD, or the presence of a pathogenic/likely pathogenic variant [4].

Genetic counseling is useful for explaining the genetic risk and how the disease is inherited to patients and their families, and for assessing the family history to identify other family members at increased risk of TAD, as well as providing psychosocial services and ethical guidance [91]. Management includes cascade genetic screening and/or imaging for TAD of patients’ family members, which are important for early prophylactic measures [1,91].

Since the evolution of TAA is mostly silent, but with fatal consequences, genetic screening for HTADs can be beneficial both for the detection of causative genetic mutations in affected patients, as well as for the identification of asymptomatic family members who are at increased risk and for guiding the optimal timing of preventive surgery in those with confirmed genetic aortopathy [1,10,11]. Genetic screening can facilitate personalized aortic care adapted to the genetic variant identified in the patient, with the aim of ameliorating the morbidity and risk of sudden premature death associated with HTADs [1,10,11]. 

In the patients included in our study, the lack of molecular testing possibilities (targeted gene or panel sequencing or WES) created difficulties regarding the genetic screening of family members of patients with HTAD. Added to this was the low compliance of some of the affected families, where the family anamnesis was performed with difficulty, and in the absence of genetic testing, monitoring was conducted through a clinical evaluation correlated with echocardiography.

In syndromic forms of HTADs, the risk of recurrence in other family members correlates with the type of monogenic inheritance (dominant or recessive) of the identified mutation. In the patients with MS (P01, P02, P03, P04, P05, P06, P07, and P08) and the patient with LDS (P09) included in our study, the risk of recurrence in the descendants of the patients is 50% (correlated with autosomal dominant inheritance), while the parents of patient P010 with ATORS have a 25% risk of having other affected children, as the disease has autosomal recessive inheritance. In the patients with MS (P07 and P08) and patient P09 with LDS in whom the family history was negative (both parents with normal phenotype), the risk of recurrence in the patients’ siblings is 1–2%, because we cannot exclude the possibility of a germinal mosaicism to parents. 

The management of patients diagnosed with Marfan syndrome (P01, P02, P03, P04, P05, P07, and P08) included continuous cardiac monitoring using biannual clinical examinations and aortic imaging (echocardiography with cardiac Z-score calculation), the aim being to avoid severe lethal complications through prophylactic surgery. Also, first-degree relatives of patients with MS were periodically evaluated clinically and by echocardiography, with the recommendation of genetic testing.

The management of patient P09 diagnosed with LDS includes biannual echocardiography and magnetic resonance angiography (MRA) or computerized tomography angiography (CTA) to evaluate the aortic root and the heart valves, as well as for the identification of the presence and/or progression of aneurysms found elsewhere in the arterial tree. The presence of scoliosis also requires orthopedic follow-up.

In patient P10 diagnosed with ATORS, management included regular cardiovascular follow-up (echocardiography, blood pressure measurements, and regular EGC and MRA and/or CT scan) to detect aortic aneurysms and making a decision regarding possible surgical interventions (aortic root replacement for aortic aneurysms and pulmonary artery reconstruction). The multidisciplinary management also included ophthalmological and orthopedic evaluations, which excluded other anomalies associated with ATORS. Also, the patient’s first-degree relatives were evaluated clinically and by echocardiography, MRA, and/or CT scan, and genetic testing was recommended. In the individuals in whom the genetic variant present in the family was detected, regular cardiovascular follow-up was also recommended.

### 3.3. Genomic Insights into HTADs

Genomic medicine and new genetic research based on a single nucleotide polymorphism (SNP) analysis and RNA expression studies will be able to contribute both to the development of targeted gene therapies and to the development of simpler and cheaper genetic tests for patients with susceptibility to HTADs or other vascular anomalies [92,93,94,95].

Future research directions involve risk stratification in relatives of patients with HTAD based on the family history, as well as a better characterization of the genetic factors involved, the discovery of new pathogenic genes, and individualization of screening algorithms [92,96].

Extensive studies such as the genome-wide association studies (GWASs) of thoracic aortic aneurysm (TAA) and dissection (TAD) will contribute to the identification of new loci/genetic risk variants, allowing the genetic architecture of HTADs to be deciphered. Based on the new findings, it will be possible to create polygenic risk scores (PRSs) that will allow the identification of individuals with a genetic susceptibility to HTADs [97].

In a GWAS that analyzed 8626 individuals with TAA/TAD included in the Million Veteran Program, Klarin et al. [97] identified 21 risk loci for TAA/TAD, 17 of which were not previously reported. The authors’ conclusion was that the obtained results demonstrate the genetic heterogeneity of TAA/TAD, showing at the same time that the genetic architecture of TAA/TAD may be similar to that of other complex, multifactorial diseases and is not inherited only through rare and highly penetrant genetic variants that cause the disease in all individuals who inherit the respective allele (rare variants with large effect sizes) [96].

We consider that the small number of cases analyzed represents a limitation of our study, and our results cannot be extended to all patients in whom we suspect a syndromic or isolated form of HTAD from the region of Moldova. One of the causes is represented by the lack of molecular testing possibilities for all patients who meet the clinical/imaging criteria for syndromic forms of HTADs, including their family members (the cost of the genetic analysis is borne by the family). Added to this are the difficulties encountered in some cases in obtaining precise information related to the family history of TAA/TAD. It is also known that the genetic variants present in different populations can vary, with some studies reporting both variants already known to be pathogenic or likely pathogenic, as well as novel genetic variants.

Based on this aspect, we consider our results to be consistent with those of other studies in the literature that included large cohorts of patients with syndromic forms of HTADs.

## 4. Materials and Methods

We analyzed a group of ten patients (including their families) with a clinical suspicion of syndromic HTADs using the records of the Cardiology Clinic at the Children’s Emergency Clinical Hospital, St. Maria Iasi, Romania. All patients came from different regions of Moldova. The diagnosis was confirmed by extensive genetic testing using next-generation sequencing (NGS) (Illumina technology, San Diego, CA, USA): a targeted gene panel using the TruSightCardio Illumina kit (Timișoara Genomic Medicine Center Laboratory, Timisoara, Romania) or whole exome sequencing—WES at a laboratory abroad (Blueprint Genetics, Espoo, Finland).

Sequencing of a panel of 174 genes was carried out in the Research Laboratory of the Timișoara Genomic Medicine Center using the Illumina TruSight Cardio Sequencing Panel kit. This panel also includes candidate genes for aortopathies and EDS: *ACTA2*, *COL3A1*, *COL5A1*, *COL5A2*, *EFEMP2*, *ELN*, *FBN1*, *FBN2*, *MYH1*, *MYLK*, *SLC2A10*, *SMADTGFB2*, *TGFB3*, *TGFBR1*, and *TGFBR2.* The NGS technique using Illumina’s MiSeq platform was employed, which allows the analysis of coding sequences from genomic DNA. In the first step, genomic DNA fragmentation was performed, followed by the amplification of coding sequences and the generation of libraries using the Illumina TruSight Cardio Sequencing Panel kit. End-to-end bioinformatics algorithms were implemented, including nitrogenous base alignment, primary filtering of low-quality reads and likely artifacts, and variant annotation, using Isis (Analysis Software) 2.5.1.3; BWA (Aligner) 0.6.1-r104-tpx; SAMtools 0.1.18 (r982:295); and Annovar (Variant Caller) 1.7. Data analysis was performed at the level of current knowledge using the following databases: UCSC (University of California Santa Cruz Genomisc Institute) Genome Browser, OMIM (Online Mendelian Inheritance in Man), and DGV (Database of Genomic Variants). All disease-causing variants reported in HGMD^®^ and ClinVar (class 1), as well as all variants with a minor allele frequency (MAF) less than 1% in the ExAc database were considered. The evaluation was focused on exons. All transmission patterns were considered, taking into account the family history and clinical information. Reported variants were correlated with the clinical phenotype. The variants were interpreted according to the American College of Medical Genetics and Genomics and Association for Molecular Pathology (ACMG/AMP) guidelines. Mendelian variants were also classified according to ACMG as pathogenic, likely pathogenic, variant of uncertain significance (VUS), likely benign, or benign.

In a single patient (P09, P.D.), the diagnosis resulted from the WES analysis performed at the Blueprint genetics laboratory. For the WES analysis, the total genomic DNA was extracted from the biological sample using a bead-based method. The quantity of DNA was assessed using a fluorometric method. After the assessment of DNA quantity, the qualified genomic DNA sample was randomly fragmented using non-contact, isothermal sonochemistry processing. A sequencing library was prepared by ligating sequencing adapters to both ends of DNA fragments. Sequencing libraries were size-selected with a bead-based method to ensure the optimal template size and amplified by polymerase chain reaction (PCR). Regions of interest (exons and intronic targets) were targeted using a hybridization-based target capture method. The quality of the completed sequencing library was controlled by ensuring the correct template size and quantity and to eliminate the presence of leftover primers and adapter–adapter dimers. Ready sequencing libraries that passed the quality control were sequenced using Illumina’s sequencing-by-synthesis method with paired-end sequencing (2 × 150 bases). The primary data analysis converting images into base calls and associated quality scores was carried out by the sequencing instrument using Illumina’s proprietary software, generating CBCL files as the final output. For bioinformatics and quality control, base-called raw sequencing data were transformed into FASTQ format using Illumina’s software (bcl2fastq). Sequence reads of each sample were mapped to the human reference genome (GRCh37/hg19). Burrows–Wheeler Aligner (BWA-MEM) software was used for read alignment. Duplicate read marking, local realignment around indels, base quality score recalibration, and variant calling were performed using GATK algorithms (Sentieon) for nDNA. Variant data were annotated using a collection of tools (VcfAnno and VEP) with a variety of public variant databases, including but not limited to gnomAD, ClinVar, and HGMD. The median sequencing depth and coverage across the target regions for the tested sample were calculated based on MQ0 aligned reads. The sequencing run included in-process reference sample(s) for quality control, which passed our thresholds for sensitivity and specificity. The patient’s sample was subjected to thorough quality control measures, including assessments for contamination and sample mix-up. Copy number variations (CNVs), defined as single exon or larger deletions or duplications (Del/Dups), were detected from the sequence data using a proprietary bioinformatics pipeline. The difference between observed and expected sequencing depth at the targeted genomic regions was calculated, and regions were divided into segments with a variable DNA copy number. The expected sequencing depth was obtained using other samples processed in the same sequence analysis as a guiding reference. The sequence data were adjusted to account for the effects of varying guanine and cytosine contents. The pathogenicity potential of the identified variants was assessed by considering the predicted consequence, the biochemical properties of the codon change, the degree of evolutionary conservation, as well as a number of reference population databases and mutation databases, such as but not limited to the 1000 Genomes Project, gnomAD, ClinVar, and HGMD Professional. For missense variants, in silico variant prediction tools such as SIFT, PolyPhen, and MutationTaster were used to assist with variant classification. In addition, the clinical relevance of any identified CNVs was evaluated by reviewing the relevant literature and databases such as the 1000 Genomes Project, Database of Genomic Variants, ExAC, gnomAD, and DECIPHER.

The pathogenicity of the identified gene variants was assessed according to the American College of Medical Genetics and Genomics and Association for Molecular Pathology (ACMG/AMP) guidelines. For the interpretation of the variants identified in patients with HTAD, we also used the HGMD Professional and ClinVar databases.

## 5. Conclusions

HTADs are conditions with an insidious evolution and are associated with a high mortality rate caused by their complications, which are often lethal, represented by TAA and TAD. These complications can be prevented by early diagnosis, periodic monitoring, and prophylactic surgical intervention.

With the advances made in genomic medicine and the development of molecular technologies based on NGS, as well as the possibility of extensive testing (targeted gene panels, WES, or GWASs), important steps have been taken in deciphering the genetic heterogeneity of HTADs correlated with phenotypic variability. In the future, new discoveries will allow the early diagnosis and personalized treatment of aortic disease, correlated with the type of genetic variant identified.

The aim of our study was to identify the genetic variants involved (already known allelic variants or new variants) in a group of patients with clinical suspicion of a syndromic form of HTAD using gene sequencing (NGS) of a gene panel (Illumina TruSight Cardio Sequencing Panel) or WES in some cases.

Our study underlines the importance of high-performance molecular genetic testing for the early diagnosis of syndromic forms of HTADs, both in patients and their family members. In the absence of genetic testing, the diagnosis could have been delayed due to the lack of clinical symptoms at the time of evaluation (taking into account the young age of some of the studied patients), especially in patients with a negative family history for HTADs.

Future research will allow the risk stratification of relatives based on family history, the identification of new allelic risk variants that will be the basis for the creation of PRSs that can be used to identify individuals at high risk for HTAD, as well as the individualization of genetic screening algorithms in affected families.

The early diagnosis of people at increased risk for HTADs requires prophylactic measures that include periodic cardiological examinations associated with aortic imaging (cardiac ultrasound—cardiac Z-score, CTA, and MRA) for the early diagnosis of TAA, reducing the risk of lethal complications through prophylactic surgical intervention.

## Figures and Tables

**Figure 1 ijms-25-11173-f001:**
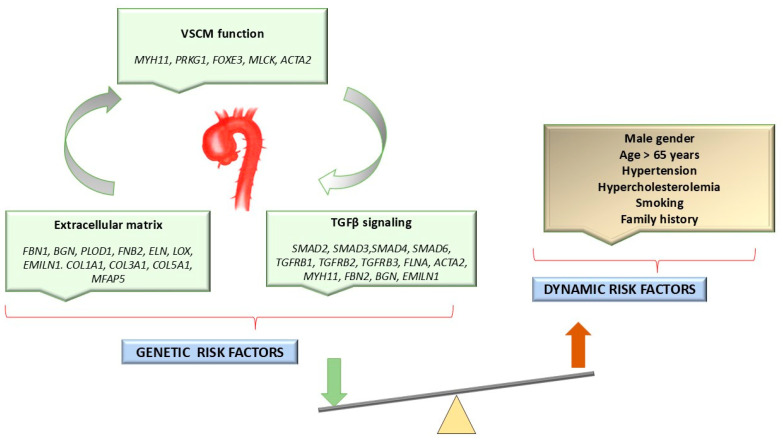
Genetic and dynamic (modifiable) risk factors for HTAD. VSCMs: vascular smooth muscle cells.

**Figure 2 ijms-25-11173-f002:**
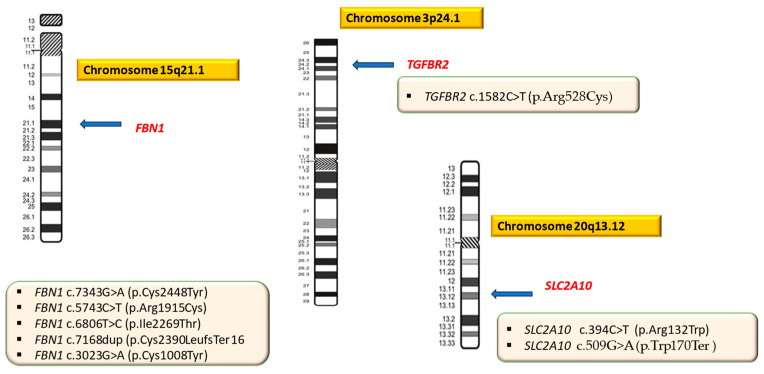
The spectrum of genetic variants detected in patients with syndromic HTAD. *FBN1*: fibrillin 1; *TGFBR2:* transforming growth factor-beta receptor, type II; *SLC2A10*: solute carrier family 2 (facilitated glucose transporter), member 10.

**Figure 3 ijms-25-11173-f003:**
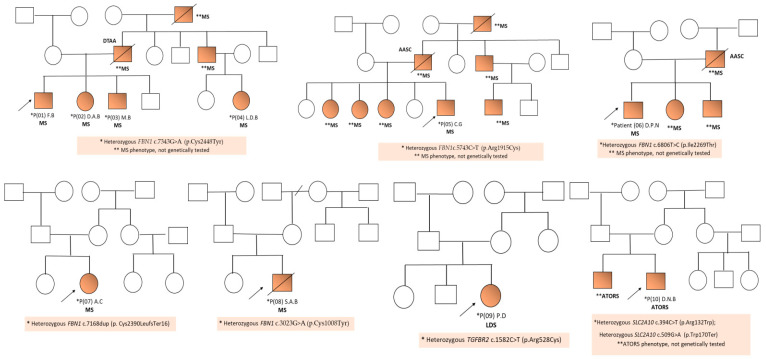
Family trees and data collected from patients with syndromic HTAD.

**Table 1 ijms-25-11173-t001:** Genetic heterogeneity in syndromic and non-syndromic HTADs [1,6,7,12,13].

Gene	Locus	OMIM	Protein	Syndrome	Inheritance Pattern	Clinical Features	References
Syndromic TAA/TAD							
*FBN1*	15q21.1	154700	Fibrillin 1	MS	AD	PE/PC, SC, PP, EL, MVP, ARD, ARH, TS, WT, and CFD	[1,6,7,12,13]
*TGFBR2*	3p24.1	190182	Transforming growth factor beta receptor 2	LDS	AD	AT, AA, hypertelorism, and bifid uvula or cleft palate	[1,6,7,12,13]
*TGFBR1*	9q22.33	609192	Transforming growth factor beta receptor 1	LDS	AD		[1,6,7,12,13]
*SMAD3*	15q22.33	603109	MADH3	LDS3	AD		[1,6,7,12,13]
*COL3A1*	2q32.2	130050	Collagen, type III, alpha-1	EDSVASC	AD	Arterial and bowel rupture; uterine rupture during pregnancy; easy bruising, thin skin with visible veins, CFD; LH and SH are minimal or absent	[1,6,7,12,13]
*SLC2A10*	20q13.12	208050	Solute carrier family 2 member 10	ATORS	AR	Tortuosity, elongation, stenosis, and aneurysms of the major arteries; LH or contractures, SH, IH; CFD (micrognathia, elongated face, high palate, beaked nose); LVH	[1,6,7,12,13]
*BGN*	Xq28	300989	Biglycan	MLS	LX	Early-onset AA and AAD; PE/PC; hypertelorism, LH or, contractures, and mild SD	[1,6,7,12,13]
*LTBP3*	11q13.1	601216	Latent TGF-b-binding protein	DASS	AR	SS with brachyolmia, HAI with almost absent enamel; MVP, ARD or other arterial aneurysms	[1,6,7,12,13]
Non-syndromic/isolated TAA/TAD							
Unknown	11q23.3-q24	607086		AAT1	AD	Medial necrosis’ or ‘Erdheim cystic medial necrosis’	[1,6,7,12,13]
Unknown	5q13-q14	607087		AAT2	AD	TAA/TAD	[1,6,7,12,13]
*MYH11*	16p13.11	132900	Myosin heavy chain 11	AAT4	AD	TAA and/or PDA	[1,6,7,12,13]
*MYLK*	3q21.1	613780	Myosin light chain kinase	AAT7	AD	TAA/TAD	[1,6,7,12,13]
*PRKG1*	10q11.23-q21.1	176894	CGMP-dependent protein kinase 1	AAT8	AD	TAA/AD	[1,6,7,12,13]
*MFAP5*	12p13.31	601103	Microfibril-associated protein 5	AAT9	AD	TAA	[1,6,7,12,13]
*LOX*	5q23.1	153455	Lysyl oxidase	AAT10	AD	TAA	[1,6,7,12,13]
*ACTA2*	10q23.31	611788	Actin alpha 2, smooth muscle (ACTA2)	AAT6	AD	TAA leading to TAD, livedo reticularis visible on arms and legs; iris flocculi; PDA; BAV	[1,6,7,12,13]
*FOXE3*	1p33	601094	Forkhead box protein E3	AAT11	AD	TAA/TAD	[1,6,7,12,13]
*MAT2A* glu344ala (E344A); (R356H) variants	2p11.2	601468	Methionine adenosyltransferase	VUS	AD	TAA/TAD	[1,6,7,12,13]
*NOTCH1*	9q34.3	190730	Neurogenic locus notch homolog protein 1	AOVD1	AD	AV calcification, stenosis and insufficiency; HLHS	[1,6,7,12,13]
*FBN1*	15q21.1	154700	Fibrillin 1	MS	AD	PE/PC, SC, PP, EL, MVP, ARD, ARH, TS, WT, and CFD	[1,6,7,12,13]

HTAD: hereditary thoracic aorta disease; TAA: thoracic aortic aneurysm; TAD: thoracic aortic dissection; MS: Marfan syndrome; LDS: Loeys–Dietz syndrome; ATORS: arterial tortuosity syndrome; EDSVASC: vascular Ehler–Danlos syndrome; MLS: Meester–Loeys syndrome; DASS: dental anomalies and short stature syndrome; OMIM: Online Inheritance in Men database; AD: autosomal dominant; AR: autosomal recessive; LX: X-linked; *FBN1*: fibrillin 1; MVP: mitral valve prolapse; AA: aortic aneurysm; AAD: acute aortic dissection; AT: arterial tortuosity; AAT6: aortic aneurysm, familial thoracic 6; ACTA2: actin alpha 2, smooth muscle; AAT1: aortic aneurysm, familial thoracic 1; AAT2: aortic aneurysm, familial thoracic 2, *MYH11*: myosin heavy chain 11; AAT4: aortic aneurysm, familial thoracic 4; PDA: patent ductus arteriosus; *MYLK*: myosin light chain kinase; *SLC2A10:* solute carrier family 2 (facilitated glucose transporter), member 10; *COL3A1:* collagen, type III, alpha-1; *SMAD3*: mothers against decapentaplegic, Drosophila, homolog of, 3 (MADH3); *TGFBR1:* transforming growth factor beta receptor 1; *PRKG1:* protein kinase, cGMP-dependent, regulatory, type 1; AAT8: aortic aneurysm, familial thoracic 8; *MFAP5*: microfibril-associated protein 5; *LOX*: lysyl oxidase; *FOXE3*: forkhead box E3; *MAT2A*: methionine adenosyltransferase II, alpha; VUS: variant of unknown significance; *NOTCH1*: Notch receptor 1; AV: aortic valve; AOVD1: aortic valve disease 1; BAV: bicuspid aortic valve; HLHS: hypoplastic left heart syndrome.; PP: pes planus; ARD: aortic root dilatation; ARH:arachnodactyly; WT: wrist and thumb sign; EL: ectopia lentis; TS: tall stature; CFD: craniofacial dysmorphism; LH: ligamentous hyperlaxity; SH: skin hyperextensibility; IH: inguinal hernias; LVH: left ventricular hypertrophy; SD: Skeletal dysplasia; SS: short stature; PE: pectus excavatum; PC: pectus carinatum; SC: scoliosis; HAI: hypoplastic amelogenesis imperfecta.

**Table 2 ijms-25-11173-t002:** Genetic heterogeneity correlates with variable phenotypes in patients with syndromic HTAD.

Patient ID	Mutation	Locus	OMIM	Transcript	Protein	Genotype	Effect of the Mutation/Pathogenicity	Variant Previously Reported	Syndrome
P01 (F.B)	*FBN1* c.7343G>A	15q21.1	134797	NM_000138.4	p.Cys2448Tyr	Hz	Missense/LP	new	MS
P02 (D.A.B)	*FBN1* c.7343G>A	15q21.1	134797	NM_000138.4	p.Cys2448Tyr	Hz	Missense/LP	new	MS
P03 (M.B)	*FBN1* c.7343G>A	15q21.1	134797	NM_000138.4	p.Cys2448Tyr	Hz	Missense/LP	new	MS
P04 (L.D.B.)	*FBN1* c.7343G>A	15q21.1	134797	NM_000138.4	p.Cys2448Tyr	Hz	Missense/LP	new	MS
P05 (C.G)	*FBN1* c.5743C>T	15q21.1	134797	NM_000138.4	p.Arg1915Cys	Hz	Missense/LP	new	MS
P06 (D.P.N)	*FBN1* c.6806T>C	15q21.1	134797	NM_000138.4	p.Ile2269Thr	Hz	Missense/LP	known	MS
P07 (A.C)	*FBN1* c.7168dup	15q21.1	134797	NM_000138.4	p. Cys2390LeufsTer16	Hz	Frameshift/LP	new	MS
P08 (S.A.B)	*FBN1* c.3023G>A	15q21.1	134797	NM_000138.4	p.Cys1008Tyr	Hz	Missense/LP	new	MS
P09 (P.D)	*TGFBR2* c.1582C>T	3p24.1	190182	NM_003242.6	p.Arg528Cys	Hz	Missense/Pathogenic	known	LDS
P10 (D.N.B)	*SLC2A10* c.394C>T	20q13.12	606145	NM_030777.3	p.Arg132Trp	Hz	Missense/Pathogenic	known	ATORS
	*SLC2A10* c.509G>A	20q13.12	606145	NM_030777.3	p.Trp170Ter	Hz	Nonsense/Pathogenic	new	ATORS

HTAD: hereditary thoracic aorta disease; LP: likely pathogenic; MS: Marfan syndrome; LDS: Loeys–Dietz syndrome; ATORS: arterial tortuosity syndrome (ATORS); *FBN1*: fibrillin 1; *TGFBR2:* transforming growth factor-beta receptor, type II; *SLC2A10*: solute carrier family 2 (facilitated glucose transporter), member 10.

**Table 3 ijms-25-11173-t003:** Clinical data of patients with syndromic HTAD.

Patient ID	Age * (m/y)	Sex M/F	Family History of AAD	Cardiac Features	Ao dil/Ao dis	Aortic Root Z Score ≥ 2	Aortic/Arterial Tortuosity	Skelethal Features	Ocular Findings	CTD Features	Other Clinical Features	SD
P01 (F.B)	12 y	M	+	MVP, MI	dilated CS	-	-	TS; PE, ARH, SC, CV, WT	-	Striae, inguinal hernia	CFD, phimosis	MS
P02 (D.A.B)	9 y	F	+	MVP, MI	dilated CS	-	-	PE, ARH, PP,	Myopic astigmatism	-	CFD	MS
P03 (M.B)	7 y	M	+	MVP, MI	AB ectasia	-	-	TS, SC, PE, ARH, PP, WT	Severe myopia	-	CFD, MC, sleep disorder	MS
P04 (L.D.B.)	12 y	F	+	MVP	-	-	-	PC, PP, ETT	Severe myopia	Inguinal hernia	CFD	MS
P05 (C.G)	9 y	M	+	MVP, TI, PI	-	-	-	TS, PE, ARH, PP, WT, SC	Myopia	Striae	CFD, Accessory spleen	MS
P06 (D.P.N)	6 y	M	+	MVP	AB ectasia	+ (Z score 3.55)	-	TS, PE, PP, ARH,	-	LH	CFD	MS
P07 (A.C)	20 y	F	-	MVP (annuloplasty), MI, AI	AB ectasia	+ (Z score 5.24)	-	TS, ARH, PC, SC, WT	Severe myopia	Striae	CFD	MS
P08 (S.A.B)	8 y	M	-	MVP, AI, MI, AH, SA	AB ectasia	+ (Z score 3.2)	-	TS, ARH, PP, WT, SC	Ectopia lentis, congenital cataract	Inguinal–scrotal hernia	CFD	MS
P09 (P.D)	4 y	F	-	TI, left AAH, PAD	-	-	-	PC, SC	-	-	CFD	LDS
P10 (D.N.B)	3 m	M	-	AAH, LVDD, AI, LVH, PAS, ATORS	-	-	+	LH	-	Inguinal–scrotal hernia, SH		ATORS

* Age (y): the age of the patients (years) at the time of diagnosis; MVP: anterior mitral valve prolapse; Ao dil: aortic dilatation; Ao dis: aortic dissection; PE: pectus excavatum; PC: pectus carinatum; CS: coronary sinus; M: male; F: female; y: years; m: month; LDS: Loeys–Dietz syndrome; ATORS: arterial tortuosity syndrome; CTD: connective tissue disorder; SD: syndrome; ARH: arachnodactyly; PP: pes planus; CV: cavus foot; WT: wrist and thumb signs; TS: tall stature; CFD: craniofacial dysmorphism; MC: macrocephaly; AB: aortic bulb; ETT: external tibial torsion; MI: mitral valve insufficiency; TI: tricuspid valve insufficiency; PI; pulmonary valve insufficiency; AI: aortic insufficiency; LH: ligamentous hyperlaxity; LVH: left ventricular hypertrophy; AH: arterial hypertension; SA: sinus arrhythmia; PAD: patent arterial duct; AAH: aortic arch hypoplasia; LVDD: left ventricular diastolic dysfunction; CoA: coarctation of the aorta; PAS: pulmonary artery stenosis; SH: skin hyperextensibility; +: present clinical sign; -: clinical sign absent.

## Data Availability

The original contributions presented in the study are included in the article; further inquiries can be directed to the corresponding authors.

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
