# Peer review of "Identification of Genetic Variants Associated with Hereditary Thoracic Aortic Diseases (HTADs) Using Next Generation Sequencing (NGS) Technology and Genotype–Phenotype Correlations"

_ijms, 2024, doi:10.3390/ijms252011173_

Round 1
Reviewer 1 Report
Comments and Suggestions for Authors
This is a very thorough and well written manuscript - well done by the authorship group. The three generation pedigree analysis and ClinVar/gNOM verifications are well done! The depth of discussion is excellent.
It is particularly refreshing to see elevation of missense mutations to a higher level of priority as a potential "genetic biomarker" of these; all of these VUS will require either computational or biologic verification through variant to protein studies. However, I enjoyed your discussion related to each of these.
A few simple suggestions:
1. The methods section must precede results section, please move this up as it distracts the reader with current placement.
2. The paper is well written and flows but is lengthy, so perhaps some editorial revision of less important components is suggested, though should be at the discretion of authors so as to not disrupt the key messaging.
3. I would advise in the conclusion or discussion a referenced discussion on why missense mutations, as you have reported, can carry important biologic weight as opposed to traditional frame-shift, deletion etc mutations.
Author Response
Dear Reviewer,Thank you very much for taking the time to review this manuscript. Please find the detailed responses below and the corresponding revisions/corrections highlighted (red color) in the re-submitted files.
- The methods section must precede results section, please move this up as it distracts the reader with current placement.
R: The Material and method section follows the Results, because we followed the IJMS journal template for articles.
- The paper is well written and flows but is lengthy, so perhaps some editorial revision of less important components is suggested, though should be at the discretion of authors so as to not disrupt the key messaging.
R: Thank you for your suggestion. Indeed, our paper is long, and although we wanted to present the heterogeneity of the genetic factors associated especially with the syndromic forms of HTAD, we consider it useful to also mention the genes associated with the isolated forms of the disease, especially since some genetic variants can cause both forms of HTAD (eg, FBN1). In addition, the interaction with other factors could explain the phenotypic expressivity detected in the syndromic forms of the disease. We also considered useful the brief description of the phenotypic manifestations associated with the 3 genetic syndromes diagnosed in the patients included in the study (Marfan, Loeys-Dietz and ATORS).
- I would advise in the conclusion or discussion a referenced discussion on why missense mutations, as you have reported, can carry important biologic weight as opposed to traditional frame-shift, deletion etc mutations.
R: Thank you for your suggestion. We mentioned in the discussion section the fact that missense mutations were most frequently detected in the analyzed patients (a single missense mutation was detected), our results being consistent with the data reported in the literature (paragraph marked with red color).
" Four of the five mutations detected in patients with MS (P01, P02, P03, P04, P05, P06 and P08) were de novo missense variants, while in one of the patients (P07, A.C.) a heterozygous frameshift mutation c.7180C>T (p.(Cys2390LeufsTer16) in the FBN1 gene was identified.
Three of the detected missense mutations affect conserved cysteine residues by replacing cysteine with another amino acid in the protein structure (p.Cys2448Tyr, p.Arg1915Cys, p.Cys1008Tyr), and in only one case a highly conserved isoleucine residue located within the cbEGF domain (p.Ile2269Thr) was affected. Although the number of patients included in our study is relatively small, we believe that we can say that the results obtained are consistent with those in the literature. Thus, the data reported so far reveal that in more than 2/3 of the cases of MS, missense mutations are detected, of which only 25% affect domains other than the cbEGF domains of fibrillin-1, in their case the cardiovascular and skeletal damage is much more severe . At the time of evaluation, only two of the patients with missense mutations (P06, aged 6 and P08, aged 8) presented AB ectasia and a cardiac Z-score >2, being considered for prophylactic aortic surgery.
Also, patient P07 (A.C, aged 20) diagnosed with heterozygous frameshift mutation c.7180C>T (p.(Cys2390LeufsTer16) in the FBN1) presented severe cardiovascular manifestations (MVP with annuloplasty, AB ectasia and cardiac Z- score: 5.24), surgical intervention being indicated to avoid potentially lethal complications."
Please check the new version of the article that we have attached. Thank you once again for the effort you put into revising our article.
Reviewer 2 Report
Comments and Suggestions for Authors
Reviewing a manuscript entitled, “The Importance of Molecular Genetic Testing for Early Diagnosis, Management and Genetic Counseling in Syndromic Hereditary Thoracic Aorta Diseases (HTAD)” By Butnariu LI, et al., this is an article focusing on the importance of molecular genetic testing in HTAD. This is a very detailed and lengthy paper, but it is difficult to understand what is new about the 10 subjects studied. Most of the manuscript is a review of HTAD.
The article title does not match the content. If the title were to be reflected, the process leading up to the diagnosis, such as how early genetic testing could have saved the patient's life, would be important, but no verification of this is included.
The authors should reconstruct the introduction in a format specific to the 10 cases of hereditary thoracic aortic disease (HTAD) that were included in the study. It is too long.
Please create a storytelling appropriate length for the introduction, including the epidemiology, causative gene, clinical symptoms, diagnostic methods, etc. of HTAD.
In the last of the introduction section, the authors mentioned “The aim of our study is to identify genetic variants associated with hereditary thoracic aortic disease (HTAD) which is characterized by an increased risk of aortic dilation and dissection, in the case of a group of patients diagnosed with syndromic forms of the disease.”. If so, the process leading up to the investigation of the family line is extremely important, but there is no description of this.
Regarding MS, which is described in the first paragraph of the discussion section, the generality of MS is described as in a review, and no new evidence is found in this manuscript. The authors should basically discuss the new evidence discovered in this study. Like the introduction section, it is too long.
The genetic mutations and clinical findings of each subject described in the discussion are the results and the authors should incorporate them into the results section. Is there any new evidence regarding these characteristics as an article?
Although the authors mentioned “Genetic counseling is useful for explaining genetic risk and how the disease is inherited to patients and their families, and for assessing family history to identify other family members at increased risk of TAD, as well as providing psychosocial services and ethical guidance. Management includes cascade genetic screening and/or imaging for TAD of patients' family members, which are important for early prophylactic measures” in the discussion section, how did the authors respond to these issues? At the very least, they need to describe how the 10 subjects were diagnosed; otherwise, it would contradict the title. Importantly, new evidence was obtained in 10 subjects. The tone is too review-like, making it difficult to understand what the manuscript is trying to convey as an original article.
And you also mentioned “Genetic screening can facilitate personalized aortic care, adapted to the genetic variant identified in the patient, with the aim of ameliorating the morbidity and risk of sudden premature death associated with HTAD.” in the discussion section. What kind of care did the authors provide to the 10 subjects?
The conclusion is clearly jumping into conclusion. The authors did not intervene in these issues at all as research, and nothing can be said from this study.
Comments on the Quality of English LanguageMinor editing of English language required.
Author Response
Dear Reviewer,
Thank you very much for taking the time to review this manuscript. Please find the detailed responses below and the corresponding revisions/corrections highlighted (red color) in the re-submitted files.
Responses:
- We changed the title to be consistent with the content of the article: "Identification of Genetic Variants Associated with Hereditary Thoracic Aortic Disease (HTAD) Using Next Generation Sequencing (NGS) Technology and Genotype-Phenotype Correlations".
- Indeed, our paper is long, and although we wanted to present the heterogeneity of the genetic factors associated, especially with the syndromic forms of HTAD, we consider it useful to also mention the genes associated with the isolated forms of the disease, especially since some genetic variants can cause both forms of HTAD (eg, FBN1). In addition, the interaction with other factors could explain the phenotypic expressivity detected in the syndromic forms of the disease. We also considered useful the brief description of the phenotypic manifestations associated with the 3 genetic syndromes diagnosed in the patients included in the study (Marfan, Loeys-Dietz and ATORS).
- We restructured the Introduction section according to your recommendation.
- At your recommendation, related to the genetic screening of patients' families, we have made some clarifications (marked in red in the text):
"In the case of the patients included in our study, the lack of molecular testing possibilities (targeted sequencing of a gene, gene panel, or WES) created difficulties regarding the genetic screening of family members of patients with HTAD. Added to this was the low compliance of some of the affected families, where the family anamnesis was done with difficulty, and in the absence of genetic testing, monitoring was done through clinical evaluation correlated with ecocardiography".
- In the results section, we presented the results obtained: both the information being presented in the text, tables 2 and 3 and figures 2 and 3. In the discussion section, we discussed the results obtained by making correlations between the detected variant and the phenotype.
Although the analyzed group is relatively small and we cannot extend our conclusions to the entire population of the region of Moldova, we have analyzed the results obtained through the lens of data from the literature.
We mentioned that most of the mutations identified in patients with syndromic HTAD were de novo, and the missense variants associated with classic Marfan syndrome were the most frequently detected (these data being similar to those in the literature).
- In the Discussion section we added a new paragraph describing the management of our patients.
- Regarding the Conclusions, we consider that the goal of our study has been achieved: the identification of genetic variants in patients with syndromic HTAD, the realization of correlations between genotype and phenotype, the interpretation of the results of genetic testing in relation to the data present in the literature, taking into account the context of difficulties related to access to molecular testing, and sometimes the lack of compliance of the affected families.
Please check the new version of the article that we have attached. Thank you once again for the effort you put into revising our article.
Reviewer 3 Report
Comments and Suggestions for Authors
The current manuscript studied the genetic variants associated with a group of patients who showed syndromic manifestations of hereditary thoracic aorta diseases (HTADs). The authors identified de novo or known mutations in FBN1, TGFBR2, and SLC2A10 genes. The authors also included comprehensive introduction and discussion combining the literature with the current study. A few minor comments:
1. The title is a bit misleading since the main findings in this manuscript are the mutations related to HTADs.
2. The authors also recognized that the sample size was very small and some patients were even from the same family, caution should be paid when generalizing the findings with these mutations.
3. Please include references for Table 1.
Author Response
Dear Reviewer,
Thank you very much for taking the time to review this manuscript. Please find the detailed responses below and the corresponding revisions/corrections highlighted (red color) in the re-submitted files.
- The title is a bit misleading since the main findings in this manuscript are the mutations related to HTADs.
R: Thank you for your observation. We changed the title to be consistent with the content of the article: "Identification of Genetic Variants Associated with Hereditary Thoracic Aortic Disease (HTAD) Using Next Generation Sequencing (NGS) Technology and Genotype-Phenotype Correlations".
- The authors also recognized that the sample size was very small and some patients were even from the same family, caution should be paid when generalizing the findings with these mutations.
R: Thank you for your observation. We mentioned this aspect as a limitation of our study (lines 721-732).
- Please include references for Table 1.
R: Thank you for your recommendation. We added the bibliography in Table 1.
Please check the new version of the article that we have attached. Thank you once again for the effort you put into revising our article.
Round 2
Reviewer 2 Report
Comments and Suggestions for Authors
This is an acceptable quality. Congrats.